# Evaluating the Influence of Musical and Monetary Rewards on Decision Making through Computational Modelling

**DOI:** 10.3390/bs14020124

**Published:** 2024-02-08

**Authors:** Grigory Kopytin, Marina Ivanova, Maria Herrojo Ruiz, Anna Shestakova

**Affiliations:** 1Institute for Cognitive Neuroscience, HSE University, 101000 Moscow, Russia; 2Department of Psychology, Goldsmiths University of London, London SE14 6NW, UK

**Keywords:** Hierarchical Gaussian Filter, reward-based learning, monetary reward, musical reward, abstract reward, probabilistic learning, decision-making behaviour

## Abstract

A central question in behavioural neuroscience is how different rewards modulate learning. While the role of monetary rewards is well-studied in decision-making research, the influence of abstract rewards like music remains poorly understood. This study investigated the dissociable effects of these two reward types on decision making. Forty participants completed two decision-making tasks, each characterised by probabilistic associations between stimuli and rewards, with probabilities changing over time to reflect environmental volatility. In each task, choices were reinforced either by monetary outcomes (win/lose) or by the endings of musical melodies (consonant/dissonant). We applied the Hierarchical Gaussian Filter, a validated hierarchical Bayesian framework, to model learning under these two conditions. Bayesian statistics provided evidence for similar learning patterns across both reward types, suggesting individuals’ similar adaptability. However, within the musical task, individual preferences for consonance over dissonance explained some aspects of learning. Specifically, correlation analyses indicated that participants more tolerant of dissonance behaved more stochastically in their belief-to-response mappings and were less likely to choose the response associated with the current prediction for a consonant ending, driven by higher volatility estimates. By contrast, participants averse to dissonance showed increased tonic volatility, leading to larger updates in reward tendency beliefs.

## 1. Introduction

Rewards, whether tangible or intangible, play a pivotal role in shaping human behaviour and driving our decisions. The neural mechanisms underlying reward processing and learning have been a focal point of neuroscientific research. Seminal early work on non-human primates using an appetitive rewarding stimulus (juice) demonstrated that dopaminergic neurons in the ventral tegmental area play a vital role in modulating reward-related activity [1]. These neurons emit a ‘teaching signal’ known as the reward prediction error (RPE). This signal indicates a discrepancy between expected and actual outcomes and facilitates learning [2]. Subsequent research expanded on this, demonstrating that dopamine prediction error responses can reflect reward magnitude and probability [3], delay [4], and preferences between different rewards [5].

Unlike the ‘primary’ rewards, which are crucial for survival and include necessities such as food, water, and sex—mainly used in animal studies, ‘secondary’ rewards represent more abstract, higher-order needs such as money, power, and positive social exchanges [6], and are used in human studies. Research by Sescousse [7] and Arsalidou [8] compared human brain responses to primary (food, sex) and secondary (monetary) rewards. They found overlapping involvement of the ventromedial prefrontal cortex, ventral striatum, amygdala, anterior insula, and mediodorsal thalamus. However, money-specific responses were more prominent in the orbitofrontal cortex, indicating its role in processing secondary rewards. Primary rewards showed stronger representation in the anterior insula, with erotic stimuli particularly activating the amygdala.

Apart from monetary rewards, humans are capable of enjoying many other abstract pleasures. Neurophysiological responses to abstract rewards have been extensively studied, ranging from social stimuli [9], to poetry [10], comedy [11] and music [12,13]. A consistent observation across these studies is the activation of the brain’s core reward regions, especially the nucleus accumbens (NAc). Social stimuli, comedy, and humour, apart from NAc activation, activate the medial prefrontal cortex. Poetry additionally engages regions related to emotional and semantic processing, such as precuneus, supramarginal gyrus and insula. Lastly, music, a rich and multifaceted stimulus, activates the NAc alongside areas related to auditory, emotional, and motor processing. Thus, while the NAc is a central brain region associated with processing pleasure, each stimulus type also activates specific brain regions corresponding to their unique cognitive, emotional, or sensory properties.

Despite the advances in the investigation of abstract pleasurable stimuli using neuroimaging, computational or psychological approaches, there is a significant gap in research concerning the role of these stimuli as reinforcement in reward-based learning. Addressing this gap could shed light on the underlying motivational aspects of these pleasures. However, several factors may have limited this area of research, such as the variable subjective value of primary rewards influenced by social contexts [14], age [15] or satiety [16]. Additionally, abstract pleasurable stimuli are further modulated by primary rewards [16,17], mood [18], and cultural and personal experiences, involving more subjective valuation and potentially leading to greater variability in individual responses.

Among abstract pleasures, music stands out as a universal and salient stimulus. It has been closely associated with human culture and history for thousands of years, as evidenced by the ancient origins of musical instruments [19,20,21]. While some studies suggest inherent biological predispositions towards certain types of music, such as a preference for consonance over dissonance [22,23,24], individual music preferences are heavily influenced by training [25] and cultural exposure and familiarity [26,27]. For instance, the preference for consonance in music is deeply embedded in Western culture [28], while countries such as Russia, Ukraine, Chile, and India each exhibit unique musical preferences compared to other countries [29]. Thus, music presents an abstract pleasure. However, its inherent subjectivity also opens up opportunities for investigating its motivational aspect, especially within the framework of reward-based learning scenarios, where this subjectivity could manifest in decision making potentially leading to poorer learning or greater variability in learning outcomes.

Recent research by Gold and colleagues [30] demonstrated how a consonance preference can guide human decision-making behaviour. They used a standard decision-making protocol to study RPEs. In their study, participants learned to make choices leading to consonant or dissonant Bach chorale endings, displaying a marked preference for consonant music. Their neuroimaging findings also linked RPEs with NAc activity, suggesting music’s potential as a motivating reward signal for learning.

While Gold’s research identified the potential of music as a reward in learning, it did not specifically investigate the role of music on modulating decision making in volatile environments, or the stochastic or exploratory nature of decision making within this context [31,32]. Stochasticity in decision making refers to the probabilistic relationship between beliefs and choices. Reinforcement learning models have explained how agents infer the expected reward magnitude or reward probability associated with actions [33]. In this context, agents often choose the action associated with the minimum expected loss. However, in the long term, it may be beneficial to deviate from ‘optimal’ behaviour and explore other actions. This deviation allows agents to gather new information and learn faster about hidden relationships, potentially including more rewarding ones. In this context, higher exploration would be associated with an increased degree of stochasticity. Agents can also exhibit noise arising from physiological or molecular processes [34]. 

In the context of the musical reward task explored by Gold et al. [30], increased stochastic behaviour would imply that participants might not consistently select the option most likely to result in a consonant ending. However, this apparent exploratory behaviour may not necessarily stem from participants’ intent to learn hidden contingencies. Instead, it could be associated with participants exhibiting more stochastic decisions due to more variable preferences regarding consonant and dissonant endings. Yet this has not been assessed empirically so far. 

This variance in decisions in a musical setting could stem from the subjective nature of musical pleasure, influenced by factors such as musical training, cultural exposure, and familiarity, and, specifically, the enjoyment of dissonance by some participants. In contrast, monetary rewards, which have been recognised as effective motivators in various studies [35], tend to promote a more predictable relationship between beliefs and decisions, as their value is more universally understood and quantifiable, motivating predictable and goal-oriented behaviour. To contribute to the literature and dissociate these two approaches to decision making, a direct comparison between learning from musical or monetary reward is necessary, thereby us expanding the work by Gold et al. [30] that, which focused on musical rewards. In addition, stochastic behaviour is best explored in volatile environments, where participants are required to continuously adapt and infer the shifting probabilistic relationships between stimuli and outcomes. This dynamic aspect is more representative of learning in the real world, and therefore the focus of monetary decision-making studies, yet it was not addressed by Gold et al. [30], given their use of stable probabilistic mappings. When decisions are made in uncertain or volatile (changing) environments, the mapping between beliefs and choices can be additionally modulated by individual estimates of volatility [36]. 

In the current study, we investigate the stochastic nature of decision making when faced with abstract rewards under volatile conditions and compare this to decision making with respect to monetary rewards, which serves as a benchmark for decision-making studies. To our knowledge, there are no studies that directly compare learning under monetary and abstract rewards. Addressing this research gap, we assess whether the abstract nature of musical pleasure leads to more stochastic choices compared to the more deterministic decision-making patterns typically associated with monetary incentives. Additionally, we evaluate whether any increased stochasticity in decision making with musical rewards is associated with musical preferences regarding what constitutes a more rewarding melody ending.

To address these questions, we designed and conducted a behavioural study using two similar probabilistic binary reward-based learning tasks. These tasks were adapted from a dynamic one-armed bandit paradigm with variable reward probabilities, following the methodology of Hein et al. [37]. Unlike the static environment previously employed to explore music as a motivational reward [30], our approach required continuous adaptation by participants to infer the shifting probabilistic relationships between stimuli and outcomes. In one task, participants selected images resulting in either a monetary gain or loss. Conversely, in the other task, their choice of image determined the nature of a Bach chorale ending—either consonant or dissonant.

We predicted that the abstract nature of musical pleasure might lead participants to demonstrate more stochastic choice behaviour when music is used as a reward, potentially resulting in slower learning rates than associated with monetary reward (Hypothesis 1). Additionally, we hypothesised that in the musical task, individuals’ increased stochasticity in decisions would correlate with their musical preference ratings (Hypothesis 2). This suggests that a greater tolerance of, or preference for, dissonant endings would manifest in behaviour where the responses deviate more from the predictions about the consonant-related choice. We tested our hypotheses using a computational modelling approach, employing the Hierarchical Gaussian Filter (HGF) [34,38]. This enabled us to estimate individual learning characteristics and belief trajectories, and to compare learning under monetary and musical reward conditions. Using robust Bayesian statistics, our analysis demonstrated that participants displayed similar learning patterns for both reward types, suggesting consistent choice behaviour irrespective of the nature of the reward. However, within the musical task, individual preferences for consonance over dissonance explained some aspects of learning, specifically, stochasticity in their belief-to-response mappings and the size of the updates in the reward tendency beliefs.

## 2. Methods

### 2.1. Participants

Forty healthy individuals (aged 18–31, 20 females and 20 males, with a mean age of 21.43 and standard error of the mean 0.46) participated in this experiment following informed consent. All participants had normal hearing and normal or corrected vision and no recent diagnosis for a mental health condition or neurological disorder. The experiment was approved by the Higher School of Economics Committee on Interuniversity Surveys and Ethical Assessment of Empirical Research. 

### 2.2. Stimuli

Stimuli material consisted of musical and visual stimuli. Musical stimuli were adapted from a previous study investigating musical reward prediction errors [30]. The stimuli were 12 four-part Bach chorales recorded with the use of a musical instrument digital interface (MIDI). Six of these stimuli were in major keys, and another six in minor keys; all were in duple metre, contained four musical phrases, eight beats, and were 25.60 s long at 75 beats per minute (bpm). Each of the 12 chorales had four versions generated by combining two timbres (harp or mandolin) and two endings (consonant or dissonant). The consonant ending was the original ending of the chorale, while in the case of a dissonant ending, each note was alternately shifted by a semitone up or down, with the soprano and tenor parts initially ascending and the alto and bass parts initially descending. Visual stimuli were two fractal figures adapted from our recent work on monetary decision making [37].

### 2.3. Experimental Design 

Participants engaged in two distinct probabilistic binary reward-based learning tasks, adapted from a one-armed bandit task with dynamically changing probabilities over time (e.g., Hein et al. [37]). One task utilised a monetary reward as a reinforcer, while the other employed music as a reward for learning.

In the monetary reward task, each trial commenced with a button press, initiating a randomly selected chorale in harp timbre. After 17 s, participants chose between two images (blue or orange), randomly presented on the left/right of the screen. The selected image probabilistically determined the reward outcome, displayed as a win (2p) or a loss (0p), along with the image itself, until the chorale concluded, signalling the end of the trial. The chorale always ended consonantly. If a participant failed to respond within two seconds, a loss outcome (0p), along with the two fractal images, was displayed on the screen until the end of the chorale. This allowed participants to discriminate whether the 0p outcome was due to a non-rewarding response or a non-response (time out).

In the musical reward task, the chorale was played in mandolin timbre. While maintaining the same timing, the chosen image probabilistically determined whether the chorale ended either in a consonant or dissonant chord. A non-response within two seconds terminated the chorale, indicating the trial’s end.

The experiment used a within-subject design so that each participant completed each task in counterbalanced order on two different days, spanning 7 to 14 days. Both tasks consisted of five different contingency mappings, with each contingency block consisting of 14 to 18 trials, amounting to a total of 80 trials per task. Such a number of trials per task block is sufficient for participants to learn the task’s statistics, as was empirically assessed through pilot testing with a separate group of volunteers. We validated our task design by confirming that the empirical probabilities governing the association between the image choice and reward outcome were closely matched to the theoretical probabilities, which were: 90/10 (probability of reward for blue *p* = 0.9; probability of reward for orange *q* = 1 − *p* = 0.1), 70/30, 50/50, 30/70 and 10/90. To ensure consistency in our within-subject study design, the order of the contingency blocks was kept identical across both tasks for each participant. This approach was crucial to ascertain that any observed differences in learning and decision making were attributable to the nature of the tasks (monetary vs. musical reward), rather than differences in the order of the contingency blocks. An illustration of the task structures is presented in Figure 1 (adapted from Gold et al. [30]).

The specific task instructions given to the participants were to learn which image was most likely to lead to the preferred outcome (the chorale ending they liked the most—in the musical reward task, or money—in the monetary reward task) on each trial and adjust their choice according to changes in the mapping between the stimulus and the outcome. Participants were informed that the contingencies would change over time, but they did not receive detailed information on the frequency of this change. The total number of points achieved in the monetary reward task were translated 1 to 1 to a ruble cash payout (on average, 89.2 and a standard error of the mean 2.27 RUB, across participants). This sum was added to a fixed amount of a 200 RUB payout. After completing the musical reward task, the participants were paid 250 RUB regardless of their choices in the task. To control for individual musical preferences, we asked participants at the end of the musical reward task to rate from 1 to 10 how much they liked (disliked) consonant and dissonant endings (separately), as well as how they liked (disliked) the sound of the instrument (mandolin). At the end of the monetary reward task, they were also asked to rate from 1 to 10 how much they liked (disliked) the sound of the instrument (harp). A conducted control analysis revealed that participants did like the sound of both instruments similarly (two-sided Bayesian Wilcoxon signed-rank test, BF_10_ = 0.18, moderate evidence for H_0_ [39]) indicating no possible confounding effects of the musical instrument type. Both tasks were programmed using PsychoPy v2020.2.10 software [40] and lasted about 40 min each.

### 2.4. General Decision-Making Performance

Using a within-subject design and well-established decision-making analysis protocols, we aimed to compare learning using monetary and musical rewards. Our general measures of decision making were the win rate, the ‘win–stay lose–shift’ metric and the run length. In the musical reward task, the winning responses were those that led to a consonant ending. In the monetary reward task, the winning outcome was a 2p win. Analysis of performance during our learning tasks using the ‘win–stay lose–shift’ [41] metric was conducted as follows. On trials with a positive outcome (points won or consonant ending) for the chosen image, we counted how often participants chose that image in the next trial. Likewise, on trials with a negative outcome (zero points won or dissonant ending), we counted how often participants avoided the chosen image in the next trial. The total number of each of these types of events was then divided by the total number of wins and losses, respectively. Win–stay and lose–shift rates were calculated for each participant in each task separately. For the run-length analysis, we calculated the average run length within each experimental contingency block for each subject and task. Run length was determined by averaging the lengths of consecutive sequences of winning responses, considering only those sequences that exceeded a single repetition. In 50/50 contingency blocks, we assessed both types of possible winning responses (choosing orange or blue fractal figures) and selected the greater of the two run lengths. This approach provides a robust measure of behavioural consistency within each experimental condition, focusing on sustained patterns of responses and effectively capturing the persistence of specific behaviours.

Given that monetary rewards often have a tangible and direct value for most individuals [42], whereas musical rewards (consonant endings) are subjective, and their value may vary between individuals based on their personal musical preferences, we hypothesised that:

**Hypothesis** **1.***Musical reward will lead to more stochastic decision-making behaviour compared to the monetary one*.

We posited additional secondary hypotheses regarding how Hypothesis 1 would be expressed in each dependent variable:

**Hypothesis** **1.1.***In the monetary reward task, participants will choose the stimulus that is more likely to result in reward outcome more consistently when compared to the musical reward task. This preference will be reflected in higher monetary win rates overall, and specifically during the non-random contingency phases (90/10 and 70/30 and the reverse), when compared to their “consonant” choices in the musical reward task*.

**Hypothesis** **1.2.***Participants will exhibit higher win–stay rates in the monetary reward task compared to the musical reward task, indicating a more consistent adherence to monetary cues. Likewise, lose–shift rates will be lower in the musical reward task compared to the monetary task because a dissonant ending might not be as aversive or negative to all participants as losing money*.

**Hypothesis** **1.3.***In the monetary reward task, participants will exhibit a higher monetary run length overall, and specifically during the non-random contingency phases (90/10 and 70/30 and the reverse), when compared to their choices in the musical reward task*.

To test our hypothesis concerning win rates, we conducted a Bayesian paired-sample *t*-test, as detailed by Rouder et al. [43]. This test evaluates the evidence supporting the alternative hypothesis over the null hypothesis, using Bayes factors (BF_10_). Bayes factors quantify the ratio between the probability of observed data under one model relative to another. For this analysis, Rouder and colleagues [43] propose selecting prior distributions for the effect size in each hypothesis, following Jeffreys [44]. The null hypothesis (H_0_) assumes no effect (δ = 0), whereas the alternative hypothesis (H_1_) uses a Cauchy distribution for a non-zero effect size. This distribution is characterised by a probability density inversely proportional σ^2^: 1/σ^2^, with σ^2^ denoting the effect’s variance in our sample (distribution of between-condition differences). Using bayesFactor toolbox in Matlab, we implemented this analysis employing the JZS (Jeffreys–Zellner–Siow) prior, a default prior for 1/σ^2^, as recommended by Rouder et al. [43].

Next, to obtain a more fine-grained assessment of learning as a function of the contingency blocks, we conducted a two-way repeated measures ANOVA, with the reward type (2 levels: monetary, musical) and contingency block (3 levels: 90/10, 70/30, 50/50) as independent variables, and the win rate as the dependent variable. This analysis specified a multivariate Cauchy prior on the effects. Mauchly’s test for sphericity showed no violations of the sphericity assumption (*p* > 0.05). The Shapiro–Wilk normality test showed that winning rates were normally distributed in all the contingency blocks and reward types (*p* > 0.05), except for the 70/30 contingency block in the musical reward task (*p* = 0.0497). In the results, we report both frequentist and Bayes factor (BF) analyses of repeated measures ANOVA for hypothesis testing. *p*-values were obtained using the Kenward–Roger method. The BF analysis of the model also used ‘JZS’ priors on the effect size under the alternative hypothesis [43,45]. The evidence for the main effect then is the ratio of the Bayes factors of the full model to a restricted model (BFratio = BFfull/BFrestricted), in which everything except the main effect is kept. We interpret Bayes factors following Wetzels and Wagenmakers [39].

To test our hypothesis regarding differences in win–stay and lose–shift behaviours between the reward conditions, we performed a paired two-sided Bayesian *t*-test [43], assessing each behaviour separately. Win–stay and lose–shift behaviours were normally distributed among the participants in each of the tasks (according to the Shapiro–Wilk normality test, *p* > 0.05).

To assess the run–length behaviour, we used the same analysis procedure as in our analysis of winning rates: a two-way repeated measures ANOVA with the reward type (2 levels: monetary, musical) and contingency block (3 levels: 90/10, 70/30, 50/50) as independent variables, and run length as the dependent variable. 

All Bayes factor analyses were conducted using the bayesFactor Toolbox by Bart Krekelberg (2022) (https://github.com/klabhub/bayesFactor, accessed on 5 September 2022) in Matlab R2021a and JASP software (JASP Team (2023). JASP (Version 0.18) (Computer software).

### 2.5. Modelling Decision-Making Behaviour with the Hierarchical Gaussian Filter (HGF)

We used the Hierarchical Gaussian Filter (HGF) [34,38] to estimate each participant’s individual learning characteristics and belief trajectories during monetary and musical binary reward learning tasks. The HGF toolbox is a freely distributed, open-source software, available in TAPAS (http://www.translationalneuromodeling.org/tapas, accessed on 5 September 2022) and has been used to investigate learning across diverse settings [37,46,47,48,49]. We used version 7.1 of the HGF toolbox.

The HGF is a Bayesian generative model of the states of an agent’s environment. In essence, the HGF is a model of perception where beliefs about the hidden states of the environment $x_{1}^{\left(k\right)},x_{2}^{\left(k\right)},\ldots ,x_{n}^{\left(k\right)}$, at trial *k*, are updated hierarchically and defined as coupled Gaussian random walks. The HGF perceptual model can then be coupled to a response (decision or observation) model that associates belief estimates to decisions, and probabilistically generates responses based on those beliefs. Note that the response model here represents a “second-order” observation, while the “first-order” observation is already embodied in the perceptual model [50]. Different combinations of HGF perceptual and response models can be applied to explain behavioural data, and model comparison techniques, such as Bayesian model selection (BMS, see below), can then be applied to select the best-performing model.

In the present study, for each of our reward tasks, we tested three alternative Hierarchical Gaussian Filter (HGF) models and two reinforcement learning models. Note that the HGF toolbox has been updated to improve the numerical implementation of the HGF model equations, which are referred to as enhanced HGF (eHGF). The input to the models were the binary time series of outcome inputs $u^{\left(k\right)}$ and responses $y^{\left(k\right)}$, where trials *k* ∈ [1, 80]. Outcome inputs were either $u^{\left(k\right)}$ = 1 if the blue image was rewarding, or $u^{\left(k\right)}$ = 0 if the orange image was rewarding. Responses were either $y^{\left(k\right)}$ = 1 if the chosen image was blue or $y^{\left(k\right)}$ = 0 if the chosen image was orange. In the following, we drop the trial index *k* for simplicity, unless otherwise stated. Missed responses were ignored by the response model, but the trial, as such, was not ignored, and filtering was suspended for this trial with the inferences on hidden states remaining, as in the previous trial.

We first chose a perceptual eHGF model with three levels, where the first level, $x_{1}^{\left(k\right)}$, denotes the true state of the input for a given trial *k*. Beliefs are represented on the second and third levels and modelled as Gaussian distributions. The second level represents the trajectory of participants’ beliefs about the tendency of the contingency between stimuli (blue/orange image) and their outcomes (rewarded or not), and the third level represents the rate of change in that tendency (volatility). Gaussian belief distributions are represented by their posterior mean ($\mu_{2},\;\mu_{3}$ for levels 2 and 3, respectively) and posterior variance ($\sigma_{2},\sigma_{3}$). This perceptual model was coupled with a unit-square sigmoid response model, and this combination constituted our first perceptual-response model (M_1_). Next, we used a perceptual 2-level eHGF model with volatility fixed to a constant level and coupled it with a unit-square sigmoid response model (M_2_). Our third hierarchical Bayesian model was the 3-level eHGF coupled with a response model where the sigmoid function depends on the expectation on log-volatility trial-by-trial (M_3_) [47]. The two tested reinforcement learning models were the Rescorla–Wagner (M_4_) [51] and Sutton K1 (M_5_) [52] models. In the Rescorla–Wagner model, adjustments to value predictions are made in accordance with a prediction error, which is scaled by a fixed learning rate. Unlike the hierarchical structure of hidden states used in HGF, this model uses a single state and has a consistent learning rate that does not change across trials. Sutton K1 model updates value predictions guided by the principle of temporal difference learning. This model emphasises the importance of future rewards, adjusting predictions based not just on immediate outcomes but also considering long-term consequences. Unlike the RW model, the Sutton K1 model incorporates a more dynamic approach, where the learning rate can vary across trials, allowing for more suitable adaptation to changing environments.

Models were then compared using the freely available MACS toolbox [53] and random effects Bayesian model selection (BMS) [54]. The log-model evidence (LME) for each of the five models in both tested conditions (musical and monetary) was calculated as the negative variational free energy under the Laplace assumption and was used as a measure of model goodness-of-fit. In both conditions, learning from reward was best explained by the same computational model, the perceptual 3-level eHGF coupled with a response model that explains decisions as a function of the estimated level of log-volatility (M_3_) (see Figure 2A). The M_3_ model exhibited exceedance probabilities of 100% in both reward conditions, and model frequencies of 87% and 85% for the monetary and musical reward tasks, respectively. This indicates that it was the unequivocally preferred model across all participants (see Figure 2B). For the detailed derivation of the perceptual model, we refer the interested reader to the papers of Mathys and colleagues [34,38]. 

In the winning model, the first level $x_{1}^{\;}$ corresponds to the binary reward outcome u. The updates on this level $\mu_{1}^{\;}$ are equivalent to the input: (1)$$\mu_{1}^{\left(k\right)}=u^{\left(k\right)}$$

The second and the third level states represent the true reward tendency of the image (blue, orange), $x_{2}^{\;}$, and the volatility or rate of change of the reward tendency, $x_{3}^{\;}$. The states $x_{2}^{\;}$ and $x_{3}^{\;}$ are continuous variables evolving as Gaussian random walks coupled through their variance. The step sizes of $x_{2}^{\;}$ and $x_{3}^{\;}$ are as follows: (2)$$x_{2}^{\left(k\right)}\sim \;N\left(x_{2}^{\left(k-1\right)},exp\left(kx_{3}+\omega_{2}\right)\right)$$
(3)$$x_{3}^{\left(k\right)}\sim \;N\left(x_{3}^{\left(k-1\right)},exp\left(\omega_{3}\right)\right)$$

In Equation (2), the parameter k (was fixed to 1 in our experiment) represents the constant degree of modulation of $x_{3}^{\;}$ on $x_{2}^{\;}$. The free parameter $\omega_{2}$ represents the tonic (time invariant) part of the variance on level 2. Therefore, larger values of $\omega_{2}$ result in more rapid belief updates, and this occurs independently of the estimated current level of volatility, $x_{3}^{\;}$. In Equation (3), the free parameter $\omega_{3}$, with exp($\omega_{3}$) sometimes termed “metavolatility”, represents how estimates of environmental volatility evolve. Here, larger values of $\omega_{3}$ articulate larger changes of the task environment. Metavolatility in our task can play a relevant role in choice behaviour. Note that in our experimental tasks, the true level of volatility in the environment was constant since the experimental block contingencies changed regularly every 14–18 trials. Participants, however, still had to estimate the volatility level, which could be over- or underestimated as a function of the task and their individual learning characteristics.

Belief updating at each level i (i = 2 and 3) and on each trial *k* is driven by prediction errors (PEs) $\delta_{i-1}^{\left(k\right)}$. These are modulated by the ratios of precision from the current level and the level below:(4)$$\Delta \mu_{i}^{k}=\mu_{i}^{\left(k\right)}-\mu_{i}^{\left(k-1\right)}\propto \frac{{\hat{\pi }}_{i-1}^{\left(k\right)}}{\pi_{i}^{\left(k\right)}}\delta_{i-1}^{\left(k\right)}$$

According to Equation (4), the posterior mean update $\Delta \mu_{i}^{k}$ is the difference between the current trial posterior mean expectation *μ_i_^(k)^* and the previous one *μ_i_^(k−^*^1)^, weighted by the prediction error of the level below, δ*_i_*_−1_*^(k)^*, and the ratio of precision terms—the prediction of precision on the level below ${\hat{\pi }}_{i-1}^{\left(k\right)}$ and the current-level precision $\pi_{i}^{\left(k\right)}$ (inverse uncertainty $\sigma_{i}^{\left(k\right)}$).

The winning HGF model was coupled with a response model first proposed by Diaconescu [47], where the belief-to-response mapping assumes the participants’ decisions to be based on their expectation of environmental volatility. 

This response model, follows a unit-square sigmoid form, as described by Mathys et al. [34]:(5)$$p(y|m,\zeta )={\left(\frac{m_{\;}^{\zeta }}{m_{\;}^{\zeta }+{\left(1-m\right)}^{\zeta }}\right)}^{y}\cdot {\left(\frac{{\left(1-m_{\;}^{\;}\right)}^{\zeta }}{m_{\;}^{\zeta }+{\left(1-m\right)}^{\zeta }}\right)}^{1-y}$$

Here, m represents the predicted probability that the next outcome will be 1. In this model, the probability of choosing a response y (0 or 1) is a function of ζ, an inverse decision noise parameter. A higher ζ implies a greater likelihood of the agent choosing the option more aligned with its current belief. Parameter ζ represents the noise or exploration in the mapping from beliefs to responses [34]. 

The decision temperature parameter ζ varies trial by trial with the prediction of the environmental volatility for the current trial (here, the current prediction is the estimate in the previous trial: *k* − 1):(6)$$\zeta^{\left(k\right)}=e^{-\mu_{3}^{\left(k-1\right)}}$$

An agent with a greater expectation of environmental volatility (smaller ζ) will display more stochastic behaviour, indicating a higher likelihood of choosing a response less consistent with their current belief about the contingency mapping. When perceiving the environment as volatile and unstable, the agent infers that contingencies change rapidly, making their predictions of current trends less reliable. This leads to a more frequent selection of the alternative response, the one with a lower probability of reward. This behaviour constitutes an exploration of the task rules. If an agent perceives that task rules are frequently changing, they might choose the less likely rewarded response from recent trials to investigate if the rule has already shifted. However, as we mentioned in the Introduction, we expected participants to exhibit a more stochastic belief-to-response mapping in the musical reward task overall, but also as a function of the individual level of asymmetry in consonant/dissonant preferences. Thus, apparent exploratory behaviour could reflect musical preferences, which we will address directly below. The trial-wise estimation of environmental volatility leads the participants to respond in close accordance with their beliefs—when they infer that the environment is stable. On the other hand, participants behave in a more exploratory manner, resulting in a noisier (less deterministic) mapping of belief-to-response probabilities when they estimate the environment to be more uncertain. In essence, the fact that the perceptual 3-level eHGF coupled with a “Volatility” response model outperformed all the other used models indicates that first, participants infer both the reward tendency and the environmental volatility and second, they dynamically incorporate these inferences into their actions (responses). 

Variational inversion of the winning model provides the trial-wise trajectories of the posterior distribution of beliefs about the states $x_{i}^{\left(k\right)}$ (i = 2,3): $\mu_{i}^{\left(k\right)}$ and $\sigma_{i}^{\left(k\right)}$. Here, $\mu_{2}^{\left(k\right)}$ and $\mu_{3}^{\left(k\right)}$ are the means, denoting the participants’ expectations on each level on trial *k*. The variances, $\sigma_{2}^{\left(k\right)}$ and $\sigma_{3}^{\left(k\right)}$, in turn, represent uncertainty on each level. An illustration of the associated belief trajectories across the total of 80 trials in the musical and monetary reward tasks for a representative participant are provided in Figure 3. Regarding our chosen prior values on model parameters, we used values from previous work [37,48], where possible. These are provided in Table 1. The selection of priors for the free parameters $\omega_{2}$ and $\omega_{3}$ was not guided by previous work, as the variational inversion algorithms diverged when using those values with our shorter time series (80 trials instead of 320 or 400 trials, as in [37,48]. Instead, in line with the recommendations of Chris Mathys and colleagues (HGF toolbox [34]), we estimated the priors on $\omega_{2}$ and $\omega_{3}$ by using an ideal model observer to analyse the input received by our participants. This approach yielded a prior mean of [−2.3, 0.7] for $\omega_{2}$ and $\omega_{3}$, respectively, and for both tasks. Our prior variance on $\omega_{2}$ and $\omega_{3}$ was set to 4 (see Table 1). The model estimates were then optimised using a quasi-Newton optimisation algorithm.

Altogether, fitting the winning model to the data obtained in the experiment allows for an individual participant’s learning characterisation by the maximum-a-posteriori estimates of the model parameters set. The computational quantities of our interest for the statistical comparison between the musical and monetary reward learning characteristics were: (1) $\omega_{2}$; (2) $\sigma_{2}$—informational uncertainty on level 2 (belief uncertainty about the reward tendency); (3) environmental uncertainty $\left[e^{\kappa \mu_{3}^{\left(k-1\right)}+\omega_{2}}\right]$; (4) $\mu_{3}$—environmental log-volatility estimates (belief about the level of the environmental volatility); (5) $\omega_{3}$; (6) $\sigma_{3}$—uncertainty on level 3 (belief uncertainty about the environmental volatility). We expected $\omega_{2}$ to be higher in the monetary reward task than in the musical one, while we anticipated comparable musical and monetary $\omega_{3}$.

Because the mapping between beliefs and responses in the winning model depended on the expectation on log-volatility, $\mu_{3}^{\left(k-1\right)}$, this parameter could capture the stochasticity or exploratory nature of decisions. Regarding $\sigma_{3}$, a trial-wise estimate of uncertainty about $\mu_{3}$, greater values would increase update steps on level 3. Accordingly, larger uncertainty $\sigma_{3}$ values could trigger more stochastic behaviour by distorting expectations or beliefs about the environment. Based on our Hypothesis 1, we predicted that learning with musical reward would be associated with higher expectation on environmental volatility estimates $\mu_{3}$. We expected $\sigma_{3}$ to be comparable between conditions.

Informational uncertainty on $x_{2}$, denoted as $\sigma_{2}$, in turn, represented uncertainty in beliefs about the tendency of the reward contingencies. This variable might influence decision making by increasing the update steps on level 2, increasing the weight that PEs have on updating beliefs on this level. Because the less certain participants are about the reward tendency (indicated by a higher $\sigma_{2}$), the slower they update their beliefs about this tendency, we expected $\sigma_{2}$ to be higher in the musical reward task than in the monetary reward task. Lastly, environmental uncertainty $\left[e^{\kappa \mu_{3}^{\left(k-1\right)}+\omega_{2}}\right]$ generally refers to unpredictability in the environment, particularly regarding stability or instability in the mapping between events and outcomes in the environment, to which an agent needs to adapt for optimal behaviour and learning. A highly volatile environment might require more adaptable learning, meaning that agents should update their beliefs more quickly when faced with discrepant information. We expected comparable musical and monetary environmental uncertainty.

To dissociate learning and behaviour with both types of reinforcing stimuli in our tasks, namely musical and monetary, we evaluated the task condition differences between monetary and musical reward learning by comparing the posterior estimates of the computational modelling parameters using a paired two-sided Bayesian Wilcoxon signed-rank test. The nonparametric test was chosen due to the non-normal distribution of each parameter in either one or both types of learning, as assessed by the Shapiro–Wilk normality test (*p* > 0.05). The test was conducted using JASP software (JASP Team (2023). JASP (Version 0.18) [Computer software]).

### 2.6. Assessing Learning in the Musical Reward Task as a Function of Musical Preferences

To assess the impact of individual musical preferences on learning outcomes in the musical reward task, we analysed participants’ scores of consonant and dissonant chorale endings. To integrate both consonant and dissonant ending scores into a single metric, we calculated a consonance dominance index (CDI). The resulting index reflects individual participants’ preferences for consonance over dissonance.
(7)$$\mathrm{Consonance}\;\mathrm{dominance}\;\mathrm{index}=\frac{\mathrm{consonance}\;\mathrm{score}\;-\;\mathrm{dissonance}\;\mathrm{score}}{\mathrm{consonance}\;\mathrm{score}\;+\;\mathrm{dissonance}\;\mathrm{score}}$$

Hypothesis 1 posited that participants in the musical reward task are expected to exhibit a more stochastic mapping between their predictions and their response compared to the monetary task. As an additional key prediction, we expected that variations in stochasticity within the musical reward task would be associated with musical preferences, such as the enjoyment of dissonance by some participants, or indifference to the type of musical ending. Specifically, we hypothesised that:

**Hypothesis** **2.**
*The CDI scores will exhibit a negative correlation with environmental volatility estimates ($\mu_{3}$) in the musical reward task, while we expect no correlation between the CDI scores and monetary $\mu_{3}$ estimates.*


Additionally, we expected that learning from musical reward would depend on CDI scores. We used $\omega_{2}$ as our learning measure, as in HGF this parameter is associated with the belief updates’ speed on $x_{2}^{\;}$ (belief about the reward tendency). Larger values of $\omega_{2}$ lead to faster belief updates irrespective of the estimated current level of volatility $x_{3}^{\;}$. We assumed that dissonant ending aversion would drive participants to learn faster, updating their beliefs towards contingency mappings more rapidly. We expected, therefore, a positive correlation between $\omega_{2}$ and the CDI score in the musical reward task, while we expected no correlation between $\omega_{2}$ and the CDI score in the monetary reward task.

We performed this analysis using both Bayesian and frequentist approaches and the BayesFactor package (version 0.9.12-4.6) in R [R Core Team (2023). R: A language and environment for statistical computing. (R Foundation for Statistical Computing, Vienna, Austria. URL https://www.R-project.org/, accessed on 5 September 2022) to assess these correlations.

## 3. Results

### 3.1. Similar Decision-Making Behaviour When Learning from Musical and Monetary Reward

To evaluate our Hypothesis 1.1—that participants will show more consistent behaviour towards a winning outcome choice in the monetary reward task compared to the musical one—we compared the means of winning responses in both tasks using the paired samples Bayesian *t*-test. The test revealed moderate evidence for H_0_ with BF_10_ = 0.20 [3], indicating a comparable performance between the two modalities; specifically, mean win rates were 0.57 (standard error of the mean, SEM, 0.014) for the monetary reward task and 0.58 (0.012) for the musical task, respectively. Further, regarding the analysis of block contingencies, a 2 × 2 Bayesian repeated measures ANOVA provided extreme evidence in support of the main effect of the Block factor (BF_ratio_ = 8.19 × 10^11^, which is the ratio between the full model with factor Block and the restricted model excluding this factor). However, for the main effect of Modality and the Block:Modality interaction, there was strong evidence in favour of the null hypothesis, with BFratios of 0.047 and 0.061, respectively. We explored the block effect more in detail using post-hoc comparisons (Bayesian *t*-test, data from both reward modalities combined). A contrast of win rates for blocks 50/50 vs. 90/10, and blocks 70/30 vs. 90/10, revealed posterior odds of BF_10_ = 9.01 × 10^7^ and BF_10_ = 1.42 × 10^6^ against the null hypothesis, respectively, indicating extreme evidence in favour of H_1_. Conversely, a comparison of blocks 50/50 vs. 70/30, revealed anecdotal evidence in favour of H_1_ (BF_10_ = 1.13). Complementing the BF analysis, frequentist statistics demonstrated a significant effect of the Block factor (*F*(2, 234) = 34.575, *p* < 0.001) but not of Modality (*F*(1, 234) = 0.067, *p* = 0.796) or of the Block:Modality interaction (*F*(2, 234) = 0.949, *p* = 0.389). These results indicate that the type of reward used does not modulate reward-based learning performance, which does not support our initial Hypothesis 1.1. The win—rate results are presented in Figure 4A.

We next analysed win–stay and lose–shift rates to assess our Hypothesis 1.2. We posited that participants would show a more consistent choice for rewarding monetary cues compared to musical consonant endings. Additionally, we hypothesised that a dissonant ending might not be as aversive or negative as a monetary loss. As a result, participants would exhibit smaller lose–shift rates when learning from musical rewards. Specifically, this suggests that after encountering a dissonant ending, participants would be less likely to switch their response compared to when they observe a trial with a monetary reward. To assess the hypothesis that win–stay behaviour for monetary rewards would be greater than for musical rewards, we conducted a Bayesian paired *t*-test. The results provided moderate evidence in favour of H_0_. Specifically, the BF_10_ was 0.20, with mean win–stay behaviours of 0.72 (SEM 0.026) for the monetary task and 0.70 (SEM 0.028) for the musical task. When assessing the hypothesis that lose–shift behaviour for musical rewards would be lower compared to monetary rewards, a Bayesian paired *t*-test revealed anecdotal evidence in favour of H_0_. The BF_10_ was 0.34, with mean lose–shift behaviours of 0.61 (SEM 0.018) for the musical task and 0.58 (SEM 0.018) for the monetary task. The switch rate results are illustrated in Figure 4B. 

To assess our hypothesis 1.3 regarding run-length behaviour, we first compared the means of run lengths in both tasks. A Bayesian *t*-test revealed moderate evidence for H_0_ with BF_10_ = 0.18, with a mean run length of 4.37 (standard error of the mean, SEM, 0.24) for the monetary reward task and 4.28 (0.27) for the musical task, respectively. Further analysis of run lengths using a 2 × 2 Bayesian repeated measures ANOVA provided extreme evidence in support of the main effect of the Block factor (BF_ratio_ = 3.36 × 10^6^). For the main effect of Modality and the Block:Modality interaction, there was strong evidence in favour of the null hypothesis, with BF_ratios_ of 0.085 and 0.032, respectively. Post-hoc comparisons (Bayesian *t*-test, data from both reward modalities combined) of run lengths for blocks 50/50 vs. 90/10, and blocks 70/30 vs. 90/10, revealed posterior odds of BF_10_ = 1.93 × 10^4^ and BF_10_ = 1.47 × 10^4^ against the null hypothesis, respectively, indicating extreme evidence in favour of H_1_. Conversely, a comparison of blocks 50/50 vs. 70/30, revealed moderate evidence in favour of H_0_ (BF_10_ = 0.15). Frequentist statistics demonstrated a significant effect of the Block factor (*F*(2, 234) = 21.597, *p* < 0.001) but not of Modality (*F*(1, 234) = 0.121, *p* = 0.728) or of the Block:Modality interaction (*F*(2, 234) = 0.118, *p* = 0.889). These results demonstrated that the type of reward used does not modulate run lengths, which does not support our Hypothesis 1.3. The run length results are illustrated in Figure 4C.

The findings indicate that the nature of the reward—whether monetary or musical—does lead to similar tendencies in win rates, switch, and run length behaviours, also suggesting that participants express an equivalent aversion to monetary loss as compared to a dissonant chorale ending.

### 3.2. Model-Based Results

To estimate individual learning characteristics and belief trajectories during monetary and musical reward learning tasks, we used the Hierarchical Gaussian Filter. In each of our learning tasks, we tested three alternative enhanced Hierarchical Gaussian Filter models and two reinforcement learning models. The models were compared using random effects Bayesian model selection [54], revealing that the perceptual 3-level eHGF, coupled with a response ‘volatility’ model [36], best explained the behavioural data in each condition (see Figure 2B). By fitting the winning model to the collected data, we characterised individual participants’ learning using maximum-a-posteriori estimates of the model parameters and dynamic belief trajectories. 

A paired two-sided Bayesian Wilcoxon signed-rank test with five chains of 1000 iterations data augmentation algorithm was performed to compare each of the six computational parameters separately between the monetary and musical reward learning tasks. This statistical analysis provided moderate [39] evidence for the null hypothesis for all the computational DVs: $\omega_{2}$, $\omega_{3}$, $\sigma_{2}$, $\sigma_{3}$, $\mu_{3}$ and environmental uncertainty (Eun_2_). The corresponding BF_10_ values were 0.19, 0.29, 0.17, 0.18, 0.19, and 0.19, respectively (see Figure 5 for computational modelling results). 

The results indicate that participants succeeded in learning in a changing environment, using both monetary and musical rewards. While computational modelling provides insights into the role of volatility and uncertainty of the environment in the stochasticity of decisions and in belief updates, no significant differences were observed between the two reward types. This suggests that the participants’ learning characteristics and belief trajectories were consistent regardless of whether the reward was monetary or musical, pointing to a common trajectory of learning and adaptation in our tasks in the face of environmental uncertainty and volatility.

### 3.3. Musical Preferences Modulate Belief Updates and Stochasticity in the Musical Reward Task

To address our Hypothesis 2—that lower CDI scores will be associated with greater environmental volatility estimates ($\mu_{3}$) in the musical reward task but not in the monetary reward task—we assessed correlations between these variables. First, following the removal of outliers, we confirmed the normality of the data. The Shapiro–Wilk test yielded *p*-values of 0.76 for musical $\mu_{3}$ and 0.54 for CDI scores, suggesting a normal distribution. We then tested the H_0_ of zero true linear correlation between these variables using Bayesian correlation analysis [56]. The correlation between the CDI scores and musical $\mu_{3}$ revealed a value BF_10_ = 2.63. This indicates that the data are approximately 2.63 times more likely to support H_1_, and therefore a nonzero correlation between musical $\mu_{3}$ and CDI scores. This BF was associated with a correlation coefficient of *r* = −0.347 (*p* = 0.033). Next, the distribution of monetary $\mu_{3}$ was also shown to be normal by the Shapiro–Wilk test (*p* = 0.053). BF analysis provided anecdotal evidence for no correlation between this variable and CDI scores, with BF_10_ = 0.42 (anecdotal evidence for H_0_; *r* = −0.099, *p* = 0.553). Of note, the correlation between musical and monetary $\mu_{3}$ was not significant, albeit there was some anecdotal support for this correlation being nonzero (BF_10_ = 1.64, anecdotal evidence for H_1_, *r* = 0.3, *p* = 0.063). 

We next assessed whether aversion to a dissonant ending (greater CDI) was associated with faster learning about the probabilistic contingencies in each condition, as expressed in the tonic volatility parameter $\omega_{2}$. After outliers were removed, we assessed the normality of the data. The Shapiro–Wilk test yielded *p*-values of 0.057 for musical $\omega_{2}$ and <0.001 for monetary $\omega_{2}$, suggesting a normal distribution for the former but not for the latter. Accordingly, in the musical condition, we tested the H_0_ of zero true linear correlation between $\omega_{2}$ and CDI scores. The BF analysis provided extreme evidence for a nonzero correlation between these variables (BF_10_ = 191.30; *r* = 0.58, *p* = 0.0001). In contrast, the Spearman rank correlation between the monetary $\omega_{2}$ and CDI values was not significant (ρ = −0.15, *p* = 0.35). In addition, the correlation between musical and monetary $\omega_{2}$ was not significant (ρ < 0.001, *p* = 1). The results of the correlation analysis are illustrated in Figure 6.

The results demonstrate that the eHGF model provides two relevant parameters, estimates of environmental volatility ($\mu_{3}$) and tonic volatility ($\omega_{2}$), that are useful in explaining the observed behaviour overall, and in the musical task, in particular. In our task involving musical rewards, these parameters are associated with the ratings of musical preferences. The correlation results suggest that a relatively higher preference for dissonance corresponds to a greater likelihood of responding in a way that deviates from current beliefs (increased stochasticity or exploratory behaviour). Second, the results indicate that the parameter modulating learning rates on level 2, $\omega_{2}$, is also associated with the consonant–dissonant ratings. Specifically, a stronger aversion to dissonance (larger CDI score) is associated with larger tonic volatility, which, in the eHGF, contributes to larger belief updates on level 2. 

## 4. Discussion

Monetary incentives, due to their universally understood and quantifiable value, have frequently been employed in human decision-making paradigms [35,37,57,58]. By contrast, the abstract and subjective nature of musical pleasure has been largely understudied in decision-making contexts. In our study, we attempted to compare learning under these two different types of stimuli. Additionally, using a computational modelling approach, we identified a distinct signature of musical reward learning associated with participants’ musical preferences.

Using two analogous decision-making protocols, a within-subject design, and the Hierarchical Gaussian Filter [34,38]—a Bayesian multi-level computational model—we found evidence for similar learning patterns in both conditions. Our cohort of forty participants demonstrated consistent learning patterns, and our Bayes factor statistical analysis [39] provided moderate evidence supporting the equivalence of learning under both musical and monetary reward conditions. Within musical reward learning, though, we found evidence of musical preferences being associated with key variables governing learning behaviour. 

On the one hand, the results did not confirm our initial hypothesis postulating more stochastic decision making and poorer learning with music as a rewarding stimulus compared to learning using monetary incentives. The hypothesis was postulated based on the evidence for differential brain processes observed between primary appetitive rewards and higher-order pleasurable stimuli [7,8,9,10,11,12,13] and consequently, the different valuation of these rewards. Yet, all these stimuli, whether appetitive or non-appetitive, can guide our behaviour in everyday life [6,59]. Monetary reward, acknowledged as the one of the most universally understood and quantifiable rewarding stimuli in modern human society [15], was used in our research as a benchmark for learning, allowing us to delineate different learning behaviours.

On the other hand, when assessing learning within the musical reward task, our hypotheses were confirmed, revealing a distinct learning signature associated with abstract musical stimuli. Specifically, consonance-over-dissonance preferences were found to modulate learning behaviour. This modulation was reflected in the association between more deterministic decisions and greater tonic volatility $\omega_{2}$ (contributing to larger steps updating level 2) in individuals with a more pronounced consonance preference. Thus, even though we did not observe differences in learning between monetary and musical rewards, our analysis determined that musical preferences could influence learning behaviour under abstract rewards.

The consistency in learning using different rewarding stimuli aligns with broader cognitive neuroscience theories that underscore the human brain’s adaptability and flexibility. The human brain is understood to possess a dynamic capacity to adjust its functional connectivity patterns in response to diverse tasks and stimuli [60]. In relation to our study, the similar learning outcomes observed under musical and monetary reward conditions might be indicative of this neural adaptability. The brain’s capacity to modulate its learning strategies, irrespective of the reward’s nature, suggests an inherent ability to optimise behaviour based on the perceived value of different stimuli. This is consistent with the concept of ‘compositional coding’, where the brain might systematically associate connectivity patterns with task components, facilitating adaptive task control [60]. On the other hand, our findings provide insight into how subjective musical pleasure modulates learning behaviour. Specifically, a higher preference for consonance over dissonance in music leads to faster learning and less stochastic decision-making behaviour. These results suggest that participants, more intrinsically motivated by consonance preferences, learn faster. We speculate that this aspect of behaviour may be associated with the neural processes processing perceived value or salience of a reward [61], rather than distinguishing between its tangible and abstract attributes. Thus, musical preferences explain some of the variance in learning behaviour under abstract rewards. However, the absence of observed differences between rewards indicates that such variance also exists in the monetary reward task, but is not captured by musical preferences and is influenced by other factors. These could include a lower need for money among some participants, or fatigue, among others. Yet, the similar learning outcomes in both tasks suggest that intrinsic stimuli can be as motivating as extrinsic monetary ones.

Reflecting on our results, it is essential to acknowledge some limitations that might have influenced our findings. The controlled laboratory environment in which participants made their choices could potentially introduce an implicit bias. Specifically, participants may have been influenced by the experimental setting, leading them to select what they believed to be the ‘expected’ choice, particularly when faced with decisions between consonant and dissonant musical endings. This could explain why they may have chosen the consonant ending more often than they might have done in a naturalistic setting without supervision. Moreover, although our sample size of forty participants is adequate to assess differences in the computations underlying decision making in similar tasks [37,55], it might not capture the full range of variability present in the wider population when completing the musical and monetary reward tasks in a within-subject design. Nevertheless, the consistency of our findings, supported by robust Bayes factor analysis, gives weight to our conclusions considering these constraints.

Future investigations could employ advanced neuroimaging techniques to probe further into the neural mechanisms underpinning musical reward processing. Functional magnetic resonance imaging (fMRI) could be utilised to examine the blood-oxygen-level-dependent responses in the NAc and other core reward regions during musical reward tasks. Comparing these responses with those elicited by monetary rewards would provide insights into the similarities and differences in neural activation patterns for these two reward types. Additionally, magnetoencephalography or electroencephalography could be employed to study the oscillatory dynamics and connectivity patterns between regions involved in reward processing. Specifically, examining the phase synchronisation or coherence between the NAc and other cortical and subcortical regions during musical and monetary reward tasks could shed light on the temporal dynamics of neural networks involved in reward-based decision making. Such detailed neuroimaging analyses would not only enhance our understanding of the neural basis of musical reward processing but also provide a comparative framework for understanding how the brain processes different types of rewards. Additionally, while our study focused on the effects of consonant and dissonant Bach chorale endings, it might be worthwhile to explore how different musical elements, such as rhythm or melody, influence decision making. This would allow for a more comprehensive understanding of the role of music in reward-based learning. 

In summary, our findings contribute to the understanding of reward-based learning processes, demonstrating that humans can adapt to learning from diverse types of rewards. Our findings suggest a degree of consistency in learning patterns across monetary and musical stimuli, thereby challenging existing assumptions within the domain of decision-making research. Notably, the study highlights the distinctive role of musical preferences in shaping learning behaviour, indicating a potential influence of intrinsic motivation on cognitive processes. However, it is imperative to acknowledge the constraints of the experimental conditions and their potential impact on participant responses. This consideration underlines the necessity for further research, ideally employing neuroimaging methods. Such future work would be central to dissociate the neural processes underlying learning from different reward types, thereby enhancing our understanding of how varying types of rewards influence cognitive and behavioural adaptations.

## Figures and Tables

**Figure 1 behavsci-14-00124-f001:**
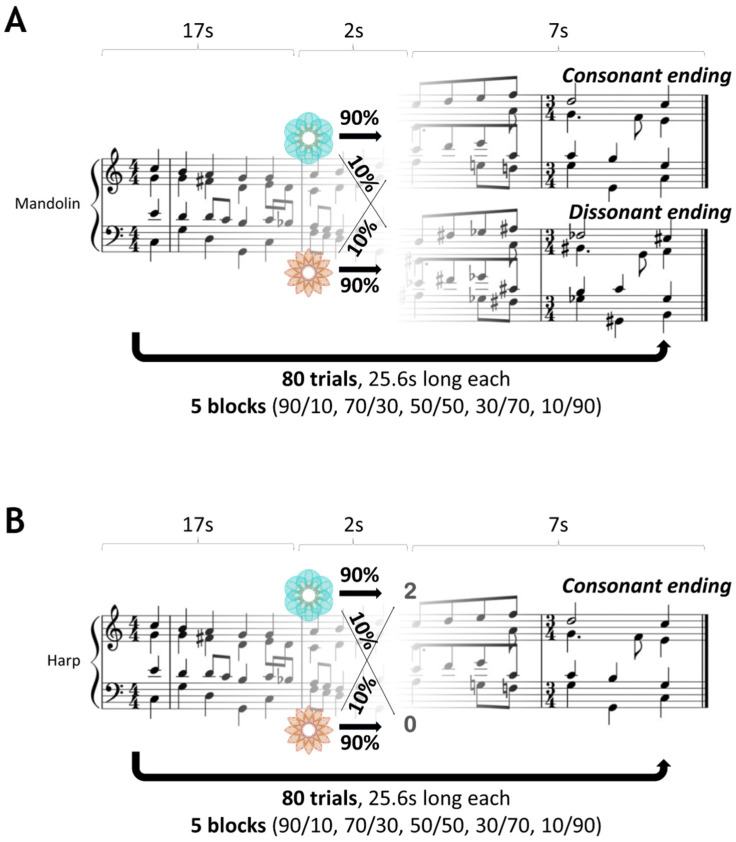
Probabilistic decision-making task structures. (**A**) Musical reward probabilistic decision-making task. Participants initiated a Bach chorale playing in a mandolin timbre by pressing a button. After 17 s, a cue prompted the participants to choose between two images. The chosen image probabilistically determined the chorale ending as either consonant or dissonant. The task had five contingency blocks (14 to 18 trials each, 80 in total) with probabilities governing the association between the image choice and chorale ending being 10% to 90%, with a step size of 20%. In the case of no response, the playing chorale aborted, moving on to the next trial. (**B**) Monetary reward probabilistic decision-making task. The same probabilistic (within-subject design) and time structure was used along with the Bach chorales, played, however, in a harp timbre. After 17 s, a cue prompted the participants to choose between two images that probabilistically determined either a win—2p or a loss—0p, when the chorale always had a consonant ending. No responses were treated as a loss—0p, without an aborting chorale being played.

**Figure 2 behavsci-14-00124-f002:**
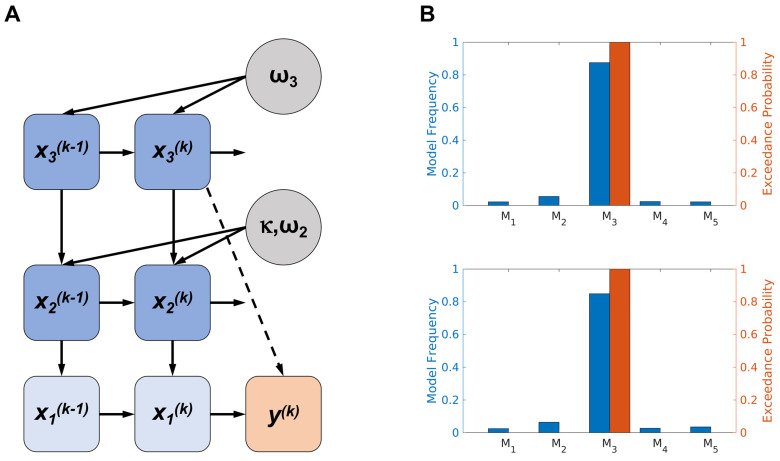
(**A**) Graphical representation of the 3-level perceptual eHGF with relevant parameters modulating each level and the associated response model. Variable $x_{1}^{\left(k\right)}$ denotes the true state of the input for a given trial *k*; $x_{2}^{\left(k\right)}$ indicates the true state of the current tendency for the mapping between blue/orange images and their outcome; and $x_{3}^{\left(k\right)}$ represents the true volatility (rate of change) of that tendency. The HGF generates trajectories of beliefs about these true states, modelled as Gaussian distributions, which are characterised by their posterior mean values ($\mu_{1},\mu_{2},\mu_{3}$) and posterior variances ($\sigma_{1},\sigma_{2},\sigma_{3}$). The parameter κ (set to 1 in our experiment) establishes the coupling strength between $x_{2}$ and $x_{3}$. The free parameters $\omega_{2}$ and $\omega_{3}$ indicate the tonic (time-invariant) component of the log volatility on each level. The response model in the winning model integrates the posterior mean of the expectation of environmental volatility with the chosen action y through a sigmoid decision rule. Here, y represents the participant’s binary response (with $y_{\;}^{\left(k\right)}$ = 1 indicating the choice of a blue image and $y_{\;}^{\left(k\right)}$ = 0 indicating the choice of an orange image). (**B**) The random-effects Bayesian model comparison confirmed that the perceptual 3-level eHGF, coupled with a response model that explains decisions as a function of the estimated level of log-volatility (M_3_), outperformed all other models utilised in the study (with M_3_ exceedance probabilities of 100% in both reward conditions and M_3_ model frequencies of 87% and 85% for the monetary and musical reward tasks, respectively). This was the case for both monetary (upper figure) and musical (lower figure) modalities. M_1_—perceptual eHGF coupled with a unit-square sigmoid response model, M_2_—perceptual 2-level eHGF model with volatility fixed to a constant level and coupled with a unit-square sigmoid response model, M_4_—Rescorla–Wagner model, M_5_—Sutton K1 model.

**Figure 3 behavsci-14-00124-f003:**
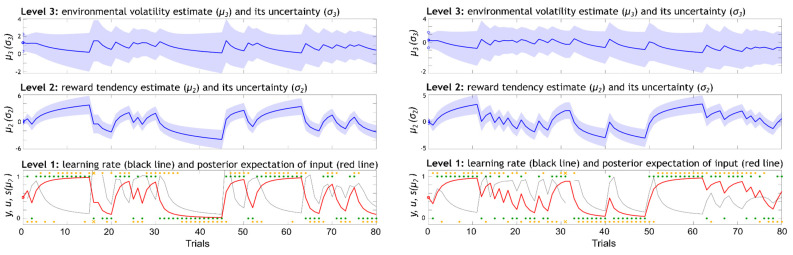
Representation of the three levels of the winning eHGF model and the associated belief trajectories across the total 80 trials in the musical (**left**) and monetary (**right**) reward learning tasks in a representative participant. The lowest level $x_{1}$ is a discrete level where the inputs u (green dots) correspond to the rewarded outcome of each trial (1 = blue, 0 = orange fractal leads to outcome); the responses y are shown in orange dots and crosses (1 = chosen blue, 0 = chosen orange, cross = no response); the red line corresponds to sigmoid transformation of $x_{2}$ and represents the posterior expectation of input; the learning rate about stimulus outcomes is given in black and represents how much new information (observing the outcomes) is used to update the model predictions. The middle level $x_{2}$ is a continuous level where beliefs $\mu_{2}\;\left(\sigma_{2}\right)$ represent the participant’s posterior mean estimate of the reward tendency and its variance. The beliefs on the level $x_{3}$, $\mu_{3}\;\left(\sigma_{3}\right)$, represent the participant’s posterior mean estimate of the environmental volatility and its variance.

**Figure 4 behavsci-14-00124-f004:**
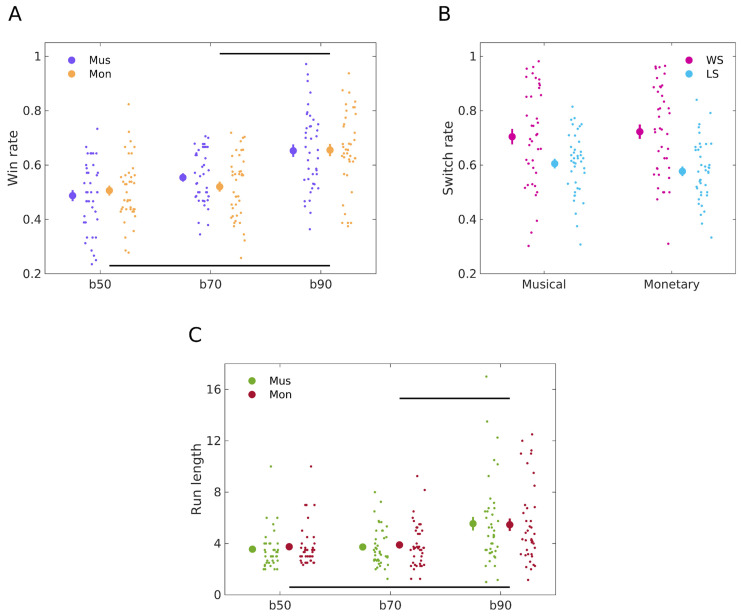
General behavioural results. (**A**) Win rates across reward types and contingency blocks. Win rates for the musical reward learning task (N = 40 participants) are depicted in purple, while orange dots represent win rates for monetary learning (N = 40 participants). Data for each task condition are illustrated using the mean (large dot) with SEM bars, with individual data points displayed to the right to showcase dispersion. A Bayesian repeated measures ANOVA (Win rate ~ Block*Modality) unveiled a significant main effect of the Block factor (*p* < 0.001) in modulating winning rates. However, the main effect of modality was not significant (*p* = 0.796), nor was the Block:Modality interaction (*p* = 0.389). Post-hoc analyses (Bayesian *t*-test, data from both modalities combined) revealed a significantly lower win rate in 50 contingency block relative to 90 (BF_10_ = 9.01 × 10^7^, extreme evidence), and in 70 relative to 90 (BF_10_ = 1.42 × 10^6^, extreme evidence), but not when comparing blocks 50 and 70 (BF_10_ = 1.13, anecdotal evidence). Differences between task conditions are indicated by horizontal lines. (**B**) Win–stay and lose–shift rates in musical and monetary reward learning. Win–stay rates are represented in magenta and lose–shift rates in blue, for both musical (N = 40 participants) and monetary (N = 40 participants) reward learning. Rates were calculated as the frequency of repeating (win–stay) or changing (lose–shift) image choices following positive or negative outcomes, respectively, normalised by total wins and losses for each task. Large dots represent the mean of each behaviour rate, with SEM bars, and individual data points are displayed to the right to illustrate dispersion. Bayesian paired *t*-test showed moderate evidence for H_0_ in win–stay behaviour, with BF_10_ of 0.20, mean behaviours of 0.72 (monetary) and 0.70 (musical), and standard errors of 0.026 and 0.028. For lose–shift behaviour, anecdotal evidence for H_0_ was found with BF_10_ of 0.34, mean behaviours of 0.61 (musical) and 0.58 (monetary), and a standard error of 0.018 for both. (**C**) Run lengths across reward types and contingency blocks. Run lengths for the musical reward learning task (N = 40 participants) are depicted in green, while red dots represent win rates for monetary learning (N = 40 participants). Data for each task conditions are illustrated using the mean (large dot) with SEM bars, with individual data points displayed to the right to showcase dispersion. A Bayesian repeated measures ANOVA (Run length ~ Block*Modality) unveiled a significant main effect of the Block factor (*p* < 0.001) in modulating run lengths. However, the main effect of modality was not significant (*p* = 0.728), nor was the Block:Modality interaction (*p* = 0.889). Post-hoc analyses (Bayesian *t*-test, data from both modalities combined) revealed a significantly lower run length in the 50 contingency block relative to 90 (BF_10_ = 1.93 × 10^4^, extreme evidence), and in 70 relative to 90 (BF_10_ = 1.47 × 10^4^, extreme evidence), but not when comparing blocks 50 and 70 (BF_10_ = 0.15, moderate evidence for H_0_). Differences between task conditions are indicated by horizontal lines.

**Figure 5 behavsci-14-00124-f005:**
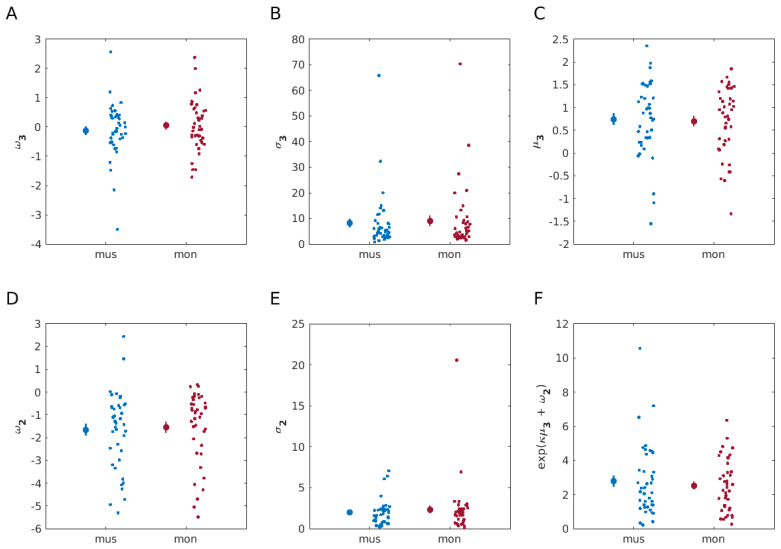
Model-based results. Computational modelling parameter comparisons between monetary (red) and musical (blue) reward modalities (N = 40 participants, each). The data for each task condition are represented by the mean (indicated by a large dot) accompanied by SEM bars. Individual data points are presented to the right to highlight the spread of the data. A paired two-sided Bayesian Wilcoxon signed-rank test, performed to compare each of the parameters separately between the reward modalities used in learning tasks, showed moderate evidence for no difference for all parameter comparisons between two learning conditions. (**A**) Tonic volatility on level 3, $\omega_{3}$, which modulates the random walk of estimates on log-volatility, $x_{3}$, BF_10_ = 0.29. (**B**) Uncertainty on level 3, $\sigma_{3}$, BF_10_ = 0.18. (**C**) Posterior mean on log-volatility, $\mu_{3}$, BF_10_ = 0.19. (**D**) Tonic volatility on level 2, $\omega_{2}$, which modulates the coupling between levels 2 and 3, and the random walk of states on level 2, $x_{2}$, BF_10_ = 0.19. (**E**) Informational uncertainty on level 2,$\;\sigma_{2}$, BF_10_ = 0.17. (**F**) Eun_2_, environmental uncertainty, BF_10_ = 0.19.

**Figure 6 behavsci-14-00124-f006:**
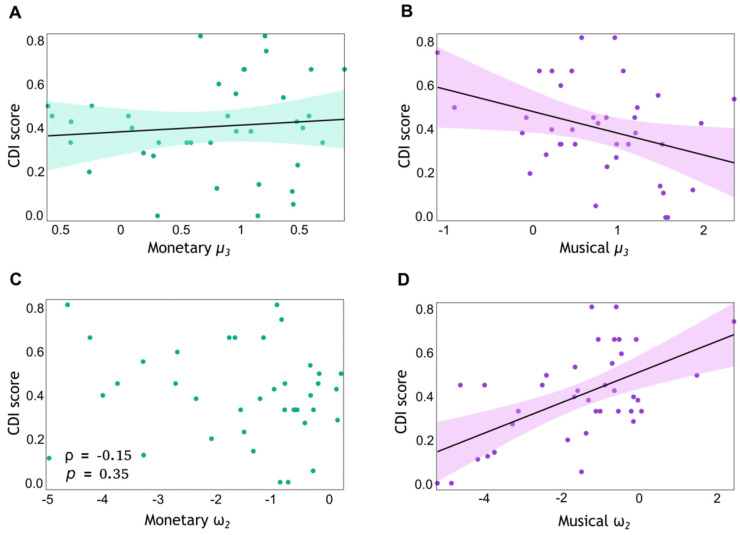
Expectation on volatility as a function of musical preferences. We conducted Bayesian and frequentist correlation analyses to investigate the relationship between CDI scores and computational modelling parameters: the posterior mean estimate of environmental volatility, averaged across trials ($\mu_{3}$), and tonic volatility $\omega_{2}$, modulating update steps on level 2. This analysis was conducted separately for monetary (green) and musical (pink) condition variables. The samples included N = 38 participants each, after outlier removal. Each point represents an individual data point: CDI score (*Y*-axis) plotted against the corresponding $\mu_{3}$ ($\omega_{2}$) value (*X*-axis). Bayesian linear correlations were conducted for variables in (**A**,**B**,**D**). Shapiro–Wilk tests confirmed the normality of the data distributions in these analyses. In panel C, a non-linear correlation analysis was conducted. The shaded areas in (**A**,**B**,**D**), represent the 95% confidence interval around the regression line. (**A**) CDI score and computational modelling parameter $\mu_{3}$ in the monetary reward task. Anecdotal evidence for H_0_ (BF_10_ = 0.42; *r* = −0.099, *p* = 0.553). (**B**) CDI score and computational modelling parameter $\mu_{3}$ in the musical reward task. The correlation revealed a value BF_10_ = 2.63, suggesting that the data are 2.63 times more likely to support H_1_ of nonzero correlation (*r* = −0.347, *p* = 0.033). (**C**) CDI score and computational modelling parameter $\omega_{2}$ in the monetary reward task. The non-significant Spearman correlation was observed (ρ = −0.15, *p* = 0.35). (**D**) CDI score and tonic volatility $\omega_{2}$ in the musical reward task. The linear correlation analysis revealed a BF_10_ = 191.30, suggesting that the data are 191.3 times more likely under H_1_ of nonzero correlation (extreme evidence; *r* = 0.577, *p* < 0.001).

**Table 1 behavsci-14-00124-t001:** Prior mean and variance of the perceptual model parameters and initial values of the belief trajectories of the winning HGF_*μ*3_ model. Parameters $\sigma_{2}^{\left(0\right)}$, $\sigma_{3}^{\left(0\right)}$, *κ* and $\mu_{3}^{\left(0\right)}$ are estimated in the log-space. The remaining parameters in the table are estimated in the natural space. In the winning HGF_*μ*3_ model, free parameters are $\omega_{2}$, $\omega_{3}$, $\mu_{3}^{\left(0\right)}$, and $\sigma_{3}^{\left(0\right)}$. See Diaconescu et al. [36] for further details on the response model and the priors. The selection of priors for the free parameters $\omega_{2}$ and $\omega_{3}$ was based on an ideal model observer to analyse the input received by our participants. See main text. The other parameters are fixed: *κ*, $\sigma_{2}^{\left(0\right)}$, $\mu_{2}^{\left(0\right)}$ [55]. The prior variances are reported in the space where the parameters are estimated.

Prior	Mean	Variance
*κ*	log(1)	0
*ω* _2_	−2.3	4
*ω* _3_	0.7	4
$\mu_{2}^{\left(0\right)}$	0	0
$\sigma_{2}^{\left(0\right)}$	log(0.1)	0
$\mu_{3}^{\left(0\right)}$	log(1)	1
$\sigma_{3}^{\left(0\right)}$	log(1)	1

## Data Availability

The data presented in this study are available on request from the corresponding author.

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
