# Peer review of "Evaluating the Influence of Musical and Monetary Rewards on Decision Making through Computational Modelling"

_behavsci, 2024, doi:10.3390/bs14020124_

Round 1

Reviewer 1 Report

Comments and Suggestions for Authors

Thank you for the opportunity to review this manuscript again, this time its revised version. The paper is now much better written on the scientific side. However, I only have the following comments:

Comment 1: The research gap has now been better explained, although I recommend emphasising more the fact where the indication of the contribution to the literature begins (e.g. by the sentence that The contribution to the literature is ...".

Comment 2: I find the 'Introduction' section slightly mixed. Despite the corrections made, I recommend adding clearly, The research gaps are such and such, and the contribution to the literature is .... .

The authors describe all this, but the lack of indication of exactly where the description of the research gaps begins and where the justification of the contribution to the literature starts makes the 'Introduction' section still slightly chaotic. Furthermore, I do not understand why, when explaining the contribution to the literature, the authors write that "we conducted a behavioural study involving 40 participants" instead of putting the information about the number of participants in the "Methods" section.

Comment 3: The numbering of the research hypotheses is unclear to me. The authors number as 1.1, 1.2, 1.3, 2. Either we indicate 2 main hypotheses and 3 secondary hypotheses, or 4 hypotheses. As it stands, it is not clear what hypothesis 1 is.

Comment 4: If the authors do not intend to separate the 'Discussion' section into two: discussion and conclusions, then the latter section should be called :Discussion and Conclusions' instead of 'Discussion'.

Reviewer 2 Report

Comments and Suggestions for Authors

The revised version is significantly improved, and all of my concerns have been properly addressed. I have no further comments.

Reviewer 3 Report

Comments and Suggestions for Authors

Referee Report for "Evaluating the Influence of Musical and Monetary Rewards on Decision Making through Computational Modelling"

General Comments: The manuscript presents a compelling study that investigates the influence of musical and monetary rewards on decision making. The authors utilized computational modeling to explore the learning process and decision-making strategies of participants under different reward conditions. The study is well-structured, and the results are effectively communicated. Overall, the manuscript provides valuable insights into the impact of diverse rewards on decision making and offers a significant contribution to the field of behavioral neuroscience. Specific Comments:

1. Introduction: The introduction provides a clear and concise overview of the study's objectives and research questions. However, it would be beneficial to include a more comprehensive review of the existing literature on the topic. Specifically, the authors could discuss previous studies that have examined the impact of various types of rewards on decision making and elucidate how their study builds upon this prior research.

2. Methods: The methods section is well-articulated and offers a detailed description of the study design, participants, and procedures. However, it would be advantageous to provide additional information on the computational modeling approach employed in the study. Specifically, the authors could elaborate on the reinforcement learning algorithm utilized and its implementation within the study.

3. Results: The results section is well-organized and effectively presents the study's findings. However, it would be beneficial to provide more detailed information on the statistical analyses employed in the study. Specifically, the authors could offer a more comprehensive explanation of the Bayesian t-tests and their application in testing the study's hypotheses.

4. Discussion: The discussion section provides a thorough interpretation of the study's findings and their implications. However, it would be advantageous to include a more comprehensive discussion of the study's limitations. Specifically, the authors could address potential confounding variables that may have influenced the results, such as individual differences in musical ability or preferences.

5. Figures and Tables: The figures and tables are well-constructed and effectively illustrate the study's findings. However, it would be beneficial to provide more detailed captions for the figures and tables, including additional information on the statistical analyses used to generate them.

6. References: The references are well-selected and provide a comprehensive overview of the existing literature on the topic. However, it would be advantageous to include more recent references, as some of the cited studies are several years old.

Overall, the manuscript presents a well-designed and well-executed study on the influence of musical and monetary rewards on decision making. The results offer valuable insights into the learning process and decision-making strategies of participants under different reward conditions. The manuscript is well-written and effectively communicates the study's findings. However, some minor revisions are needed to enhance the clarity and completeness of the manuscript.

Comments on the Quality of English Language

The quality of English language in the manuscript is high. However, some minor improvements could be made to further enhance the manuscript's clarity and impact.

Round 2

Reviewer 1 Report

Comments and Suggestions for Authors

Thank you for the opportunity to review this manuscript again. Most of my recommendations were taken into account by the authors.

I have only one comment concerning the correlation between variables. Can you explain why do you apply the Pearson coefficient to check the correlation? In the previous review, I did not draw attention to this aspect because there were so many shortcomings that I forgot to mention it in the comments.

I ask because this coefficient is used under the assumption that the relationship is linear. It is evident from the graphs that linearity is not the case here. Therefore, in this case, there is no basis for using this coefficient if linearity is absent. In the absence of linearity, using Pearson's linear correlation coefficient is useless. Have the authors checked the conditions for linearity?

Round 3

Reviewer 1 Report

Comments and Suggestions for Authors

Thank you for the opportunity to review this manuscript again.

I've read the explanation of the authors carefully. The justification was clearly presented. I am satisfied. I believe that, as things stand, the article is publishable and I congratulate the authors.